# The Potential of Co-Evolution and Interactions of Gut Bacteria–Phages in Bamboo-Eating Pandas: Insights from Dietary Preference-Based Metagenomic Analysis

**DOI:** 10.3390/microorganisms12040713

**Published:** 2024-03-31

**Authors:** Mingyue Zhang, Yanan Zhou, Xinyuan Cui, Lifeng Zhu

**Affiliations:** College of Life Sciences, Nanjing Normal University, Nanjing 210098, China; zhangmingyueaaa@163.com (M.Z.); jsz9900y@163.com (Y.Z.); cuixinyuanwj@163.com (X.C.)

**Keywords:** bamboo-eating pandas, gut microbiomes, phage, metagenomes, co-evolution, conservation

## Abstract

Bacteria and phages are two of the most abundant biological entities in the gut microbiome, and diet and host phylogeny are two of the most critical factors influencing the gut microbiome. A stable gut bacterial community plays a pivotal role in the host’s physiological development and immune health. A phage is a virus that directly infects bacteria, and phages’ close associations and interactions with bacteria are essential for maintaining the stability of the gut bacterial community and the entire microbial ecosystem. Here, we utilized 99 published metagenomic datasets from 38 mammalian species to investigate the relationship (diversity and composition) and potential interactions between gut bacterial and phage communities and the impact of diet and phylogeny on these communities. Our results highlight the co-evolutionary potential of bacterial–phage interactions within the mammalian gut. We observed a higher alpha diversity in gut bacteria than in phages and identified positive correlations between bacterial and phage compositions. Furthermore, our study revealed the significant influence of diet and phylogeny on mammalian gut bacterial and phage communities. We discovered that the impact of dietary factors on these communities was more pronounced than that of phylogenetic factors at the order level. In contrast, phylogenetic characteristics had a more substantial influence at the family level. The similar omnivorous dietary preference and closer phylogenetic relationship (family Ursidae) may contribute to the similarity of gut bacterial and phage communities between captive giant panda populations (GPCD and GPYA) and omnivorous animals (OC; including Sun bear, brown bear, and Asian black bear). This study employed co-occurrence microbial network analysis to reveal the potential interaction patterns between bacteria and phages. Compared to other mammalian groups (carnivores, herbivores, and omnivores), the gut bacterial and phage communities of bamboo-eating species (giant pandas and red pandas) exhibited a higher level of interaction. Additionally, keystone species and modular analysis showed the potential role of phages in driving and maintaining the interaction patterns between bacteria and phages in captive giant pandas. In sum, gaining a comprehensive understanding of the interaction between the gut microbiota and phages in mammals is of great significance, which is of great value in promoting healthy and sustainable mammals and may provide valuable insights into the conservation of wildlife populations, especially endangered animal species.

## 1. Introduction

The host gut is a complex microbial ecosystem comprising prokaryotic microorganisms (bacteria and archaea), eukaryotic microorganisms (including fungi, nematodes, and protozoa), and viruses, collectively referred to as the gut microbiome [1,2,3]. To date, the critical roles of gut bacterial communities in physiological development, dietary digestion, and immune health of the host have been well described [4,5,6,7,8,9,10,11]. Phages (bacteriophages) are viruses that infect prokaryotic microorganisms (bacteria and archaea) and represent the most abundant component within the virome and microbiome [12,13,14,15]. With the development and application of high-throughput metagenomic techniques [16], the understanding of phages has been newly propelled and deepened [17,18]. Like bacteria, phages are also resident members of the gut microbiota [19], with approximately ten phage particles parasitizing each bacterial cell [20,21]. Phages can be categorized into two types: virulent phages and temperate phages [13,22]. The former can directly enter the lytic cycle and lyse the host bacteria through self-replication. At the same time, the latter can also choose another infection strategy, the lysogeny cycle, in the latent phase of which temperate phages could integrate their genomes into the chromosome of the host bacteria to form a stable complex [23,24].

In addition, exploring the factors influencing gut microbiota is a fundamental research topic in microbial ecology [25]. It has been reported that diet and phylogeny are the two most important factors influencing the bacterial community in the mammalian gut [26,27,28]. Similarly, these factors also affect the host gut phage community [29,30], and there is evidence suggesting that diet also impacts the gut phage community and may lead to more persistent changes in phage composition [31,32,33]. However, when studying individual bacteria or phage communities in isolation, their relationships and interactions are often overlooked. Bacteria and phages constitute the most abundant biological entities on Earth [34,35], and both have complex interactions and the potential for rapid co-evolution [36]. Virulent phages rapidly proliferate and destroy host bacteria through the lytic cycle, enabling them to spread quickly within the host [37,38]. Temperate phages choose the lysogenic cycle, stably coexist within the host bacteria through horizontal gene transfer, and replicate and survive, utilizing the host’s survival advantage [39,40]. Horizontal gene transfer (HGT) facilitates the co-evolution of bacteria and phages [41]. Coevolving phages can increase bacterial diversity by various resistance patterns [36,42] and have the most pronounced effects on bacterial pathogens [41]. For instance, phages can enhance the beneficial characteristics and infectivity of bacterial pathogens (e.g., *Escherichia coli*, *Vibrio cholerae*, *Pseudomonas aeruginosa*, *Salmonella enterica*, *Shigella*, etc.) by transferring virulence factors through HGT [43,44,45], thereby further driving the evolution of bacterial pathogens. Moreover, co-evolution and interactions between bacteria and phages are pivotal in maintaining gut microbial diversity [46,47]. More diverse gut bacteria can potentially increase the genomic diversity of co-evolving phages, facilitating their infection of a broader range of host bacteria [46]. Simultaneously, HGT driven by phages also contributes to gut bacteria heterogeneity, thereby strengthening coevolution’s effects. 

Network analysis, as an excellent microbiome research tool, has been widely employed to gain deeper insights into the intricate relationships that shape the dynamics of microbial communities among different microbial ecosystems [48,49,50,51,52]. Additionally, network analysis assists in identifying keystone species, functional modules, and the importance of ecological niches [4,46,53]. Through in-depth investigations into the interactions and functional composition patterns of microbial communities [54,55,56], we can further explore the mechanisms underlying the functioning of the gut microbiota and its implications for host health. Network analysis has been employed to investigate the functions of fungi and protists in the giant panda gut microbiome and antibiotic resistome, yielding valuable insights, particularly highlighting the pivotal roles played by protists within the network modules [4]. Similarly, Cui et al. demonstrated variances in the topological characteristics of gut microbial networks between captive and wild giant panda populations through network module and keystone species analyses. The findings indicated that the wild giant panda [52] gut microbiome exhibits higher complexity, stability, and resilience to external influences. Cohesion, an innovative method for quantifying connectivity within microbial community networks, has gained widespread acceptance in assessing the relationships and interactions among diverse microorganisms in interdomain ecological networks. Recent studies have revealed that cohesion effectively unveils interactions (cooperation/competition levels), stability, and complexity within bacteria–fungi interdomain networks in soil and composting ecosystems [57,58]. The application of these network analysis methods presents an opportunity to attain a more profound and comprehensive understanding of the structure and function of microbial communities. It serves to further enrich our grasp of the intricate interactions among microorganisms within ecosystems.

The giant panda, a specialized bamboo eater, stands as the flagship species for global biodiversity conservation. Gut bacteria play an important role in the nutrition, development, and immunity of giant pandas, and are also affected by various external environmental factors [52]. Similarly, dietary alterations have been found to influence the diversity of gut phages in giant pandas. It was also shown that the gut phages of the giant panda are predominantly dominated by Caudovirales and Enterobacteriaceae phages, with Caudovirales displaying highly genetic diversity [59,60,61]. Furthermore, giant pandas exhibit a higher diversity and abundance of phage communities in their gut compared to other related species such as red pandas and bears [60]. An in-depth investigation of gut bacteria and phages is essential for delving into the molecular evolution of mammalian bacterial–phage interactions and for safeguarding the survival and well-being of endangered wildlife, exemplified by species like the giant panda. Furthermore, positive correlations between gut bacteria and phages in diversity and composition have been observed in humans [62] and non-human primates [29], respectively. However, the applicability of this bacterial–phage theoretical relationship to a broader range of mammalian taxa remains to be further explored. Here, we focused on bamboo-eating pandas with diverse geographical distributions as the primary study subjects, concurrently comparing them with carnivores, herbivores, and omnivores, to investigate the diversity, composition, and potential interactions of bacterial and phage communities in the mammalian gut. We aim to address the following issues: (1) determine the relationship between mammalian gut bacterial and phage communities in diversity and composition; (2) identify the influence and extent of dietary and phylogenetic factors on the host gut bacterial and phage communities; and (3) evaluate the potential connections and interactions between bacterial and phage communities in the mammalian gut. A deeper understanding of the structural composition and interactions between bacteria and phages can contribute to a comprehensive comprehension of their co-evolutionary processes. Moreover, it can provide unique insights for wildlife conservation, particularly the endangered wild animals.

## 2. Materials and Methods

### 2.1. Data Collection

This study collected 99 published metagenomes (raw data) from 38 mammalian species. These species include carnivores, omnivores, herbivores, as well as bamboo-eating species (giant panda and red panda). Among them, 52 metagenomes were obtained from giant pandas belonging to five different geographic populations: 7 (GPCD) were from a captive population in Chengdu [63], 10 (GPYA) from a captive population in Ya’an [64], 9 (GPQIN) from a wild population in the Qinling mountains [65], 7 (GPQIO) from a wild population in the Qionglai mountains [64], and 19 (GPXXL) from a wild population in the Xiaoxiangling mountains [66]. Similarly, six metagenomes (RP) of bamboo-eating red pandas were also from the Xiaoxiangling mountains [66]. Furthermore, we integrated an additional set of 41 metagenomes [67], including 19 from carnivores (CA), 12 from herbivores (HE), and 10 from omnivores (OC). Detailed information on the sample groupings and species data is provided in Appendix A.

### 2.2. Metagenomic Analyses and Bioinformatics Analysis

Quality Control using Trimmomatic: Trimmomatic [68] was employed to conduct quality control on raw reads from 99 metagenomes, filtering and removing reads less than 50 bp in length, reads with degenerate bases (N’s), and all duplicate sequences (where the initial 20 nucleotides were identical, sharing an overall identity similarity of >97% throughout the length of the shortest read). Alignment with BWA-MEM: The bioinformatics tool BWA-MEM was used to align the sequence data, identifying and removing contamination sequences originating from the putative host [69]. Megahit Assembly and Salmon Quality Control: Megahit [70] was employed for the assembly of clean reads into contigs (≥500 bp), and Salmon [71] was applied to perform quality control on the contigs, discarding those with coverage below 60%. Gene Prediction with Prodigal: Gene prediction was carried out on the metagenomic contigs using the meta mode of Prodigal [72], generating gene files. Gene Clustering with Cd-hit: Cd-hit [73] was then employed for gene clustering, constructing non-redundant (NR) gene sets with an identity threshold of 95% and an overlap threshold of 90%. Mapping and Abundance Calculation using Salmon: Salmon [71] was utilized for mapping clean reads to the reference non-redundant (NR) gene profile and calculating transcripts per million (TPM) to determine the abundance of unigenes. BLAST Analysis with Diamond: Finally, a BLAST analysis of those unigenes against the NCBI-NR database was performed using Diamond [74], and the TPM for each taxonomic group (bacteria and phages) was obtained through our customized program.

The alpha diversity of gut bacterial and phage communities among different groups was quantified using the Shannon index, which measures species diversity within a microbial community. The Bray–Curtis dissimilarity distance, a metric commonly used in ecological studies, was employed to quantify β-diversity, indicating the compositional differences between communities [75]. NMDS (non-metric multidimensional scaling) analysis, a technique that visualizes the similarity/dissimilarity of samples in a multidimensional space, was conducted using the Vegan (Version: 2.6-4) package [76] based on Bray–Curtis dissimilarity matrices [77]. This analysis aimed to reveal potential dissimilar clusters of gut bacterial and phage communities across different groups [78]. To assess the statistical significance of the impact of dietary and phylogenetic factors on the compositional variations of host gut bacterial and phage communities, the Adonis test was applied [79]. This test is particularly suitable for analyzing multivariate data and determining the influence of categorical variables on community composition. Furthermore, Sørensen’s dissimilarity was calculated using the vegdist function in the “Vegan” R package (Version: 2.6-4) [80] to examine the relationship between the bacterial community and phage community among different groups. Sørensen’s index measures the community similarity between two samples based on the presence or absence data of microbial species, providing insights into the overall community structure and composition, as described previously [29].

### 2.3. Co-Occurrence Network Analysis

In each group, the top 1000 abundant OTUs (operational taxonomic units) in bacterial communities, along with all OTUs in phage communities, were selected to conduct network analyses encompassing bacteria, phages, and bacterial–phage interdomain community. Robust correlations with Spearman’s correlation coefficients (|ρ|  ≥  0.6) and statistically significant *p*-values ≤ 0.01 (FDR-corrected details provided in Appendix A) were used to construct networks using the “Picante” R package (Version: 1.8.2) [81,82]. Subsequently, network visualization and module analysis were performed using Gephi v.0.9.2 platform [83]. Node-level topological features, including degree, betweenness centrality, closeness centrality, and eigenvector centrality values, were calculated for each network using the “Igraph” R package (Version: 2.0.3) [84]. The nonparametric Mann–Whitney U test was employed to assess the statistical significance of differences in node-level attributes measured across different taxa [85]. Newman’s method was used to calculate modularity values between 0 and 1, which can be utilized to indicate the extent to which a network can be divided into modules [86]. The Z-score value (the sum of degree and closeness centrality values) can be utilized to identify keystone taxa in the network, as described in previous studies [87], and all nodes with top 10 Z-score values were selected as keystone species for each network. Cohesion can be employed as a metric for assessing microbial interactions [88]. In accordance with the approach proposed by Herren and McMahon [89], cohesion is calculated by multiplying the abundance of each taxon in a given sample by its associated connectivity value and subsequently summing the products across all taxa. Positive and negative cohesion values are obtained based on the positive and negative correlations between taxa, respectively. The total cohesion value is the sum of positive cohesion and the absolute value of negative cohesion [90], providing a comprehensive measure of microbial interactions within the given sample.

## 3. Results

### 3.1. Diversity Analysis of Gut Bacterial and Phage Communities

This study analyzed 99 published metagenomes derived from 38 mammalian species with different dietary preferences (carnivorous, herbivorous, omnivorous, and bamboo-eating) (Appendix A). Based on the Shannon index, α-diversity analysis showed that the alpha diversity of the gut bacterial community was significantly higher than that of the phage community in CA (*p* < 0.001), HE (*p* < 0.001), GPQIN (*p* < 0.05), GPQIO (*p* < 0.01), GPXXL (*p* < 0.001), and RP (*p* < 0.01) (Figure 1a). Beta diversity analysis using Bray–Curtis dissimilarity indicated significant distinctions between gut bacterial and phage communities within each group (Figure 1b). Further NMDS analysis elucidated distinct clustering patterns among gut bacterial groups (Figure 1c) and phage communities (Figure 1d). In contrast to other groups, it was observed that samples from captive giant pandas (GPCD and GPYA) and OC exhibited a closer distance in both gut bacterial and phage communities. In gut bacterial communities (Figure 1c), wild bamboo-eating species (GPQIN, GPQIO, GPXXL, and RP) displayed a more similar composition, while both HE and CA exhibited more independent clustering patterns when compared to the other groups. The NMDS analysis of phage communities (Figure 1d) showed no evident spatial segregation patterns among wild bamboo-eating species (GPQIN, GPQIO, GPXXL, and RP), CA, and HE, but they all exhibited a distinct separation trend from the sample points of captive giant pandas (GPCD and GPYA) and OC along the *x*-axis (NMDS1 axis). These observations were further supported by hierarchical clustering analysis based on bacterial and phage compositions (Appendix A).

### 3.2. Compositions of Gut Bacterial and Phage Communities

In this study, Adonis analysis was utilized to examine the impacts of diet and phylogeny on gut bacterial and phage communities in various mammalian species (Table 1). Our results demonstrate that diet and phylogeny significantly influenced (*p* = 0.001) the composition of gut bacterial and phage communities. Furthermore, phylogenetic factors at the family level were found to be more significant contributors to dissimilarity in bacterial (*R*^2^ = 0.7153) and phage (*R*^2^ = 0.44625) communities in the gut of different animals, outweighing the influence of dietary factors (bacteria: *R*^2^ = 0.45537; phages: *R*^2^ = 0.16211). 

To further investigate the impact of diet and phylogeny on dissimilarity, we analyzed the composition of gut bacterial and phage communities in each group. Across all groups, Proteobacteria, Firmicutes, and Bacteroidetes were the major contributors to the gut bacterial community (Figure 2a). Specifically, both Firmicutes and Bacteroidetes were dominant in CA and HE, and Firmicutes constituted an absolute majority in GPQIN, while Proteobacteria exhibited significant enrichment in the OC and other bamboo-eating panda populations (GPCD, GPYA, GPQIO, GPXXL, and RP). In the gut phage community, Uroviricota (Figure 2b) and Caudovirales (Appendix A) were dominant at the phylum and order levels across all groups. At the family level of bacteria (Figure 2c), compared to the other groups, Bacteroidaceae exhibited significant enrichment in CA (32.48%), Bacteroidales_norank (11.24%), and Oscillospiraceae (11.03%) displayed greater abundance in HE, and Clostridiaceae constituted a substantial proportion in GPQIN (44.29%). Enterobacteriaceae were found to be dominant in the OC (34.71%) and captive giant pandas (GPCD: 60.27% and GPYA: 54.52%), while Pseudomonadaceae dominated the other wild bamboo-eating species (GPQIO: 18.56%; GPXXL: 46.17%; and RP: 61.42%). Additionally, at the family level of gut phage communities (Figure 2d), Podoviridae exhibited a dominant prevalence in both CA (89.53%) and HE (39.37%), Drexlerviridae demonstrated absolute dominance in GPQIN (74.83%), and Siphoviridae and Myoviridae constituted the major components in OC and other bamboo-eating species (GPCD, GPYA, GPQIO, GPXXL, and RP). In addition, we further investigated the relationships between bacterial and phage communities in composition within the gut ecosystem and observed a general positive correlation between their composition in all examined samples and groups (Figure 2e–i).

### 3.3. Co-Occurrence Microbial Networks Analysis of Gut Bacterial–Phage, Bacteria, and Phage Communities

We analyzed the co-occurrence microbial networks of bacterial, phage, and bacterial–phage interdomain communities in each group (Figure 3 and Appendix A). Our findings demonstrated that bamboo-eating species (GPCD, GPYA, GPQIN, GPQIO, GPXXL, and RP) and OC shared more similar composition patterns across these microbial networks (Figure 4a–c). Specifically, Proteobacteria, Firmicutes, Siphoviridae, and Myoviridae were more abundant in their bacterial–phage interdomain networks. Proteobacteria and Firmicutes were identified as the primary constituents in their gut bacteria networks. In their phage networks, GPQIN displayed a distinct pattern characterized by a substantial abundance of Drexlerviridae, while Siphoviridae and Myoviridae dominated in OC and other bamboo-eating panda populations (GPCD, GPYA, GPQIO, GPXXL, and RP). In addition, CA and HE exhibited pronounced similarities in their network compositions (Figure 4a–c). Bacteroidetes and Firmicutes were identified as dominant taxa in their bacterial networks, while Podoviridae constituted the most abundant component in their phage networks. Moreover, Bacteroidetes, Firmicutes, and Podoviridae emerged as prevailing taxa within their bacteria–phage interdomain networks.

Different nodes serve distinct topological roles within their network, exerting varying degrees of impact on its operation and functionality. To precisely evaluate the topological importance of each node, we utilized a high Z-score as a metric for identifying keystone species in co-occurrence microbial networks (Figure 4d–f). In bacterial networks (Figure 4d and Appendix A), it was found that Firmicutes constituted the majority of keystone species in CA (90%), GPQIN (100%), and RP (100%), Bacteroidetes dominated in HE (60%), while in OC (90%), GPCD (100%), GPYA (70%), GPQIO (100%), and GPXXL (100%), the prevailing keystone species were assigned to Proteobacteria. In phage networks (Figure 4e and Appendix A), the keystone species in the GPXXL primarily belong to Autographiviridae (60%), whereas in other groups, the most of keystone species were predominantly classified into Myoviridae (CA: 69.70%; HE: 30.00%; OC: 11.32%; GPCD: 49.28%; GPYA: 50.00%; GPQIN: 72.06%; GPQIO: 53.16%; and RP: 37.29%) and Siphoviridae (CA: 21.21%; HE: 30.00%; OC: 59.55%; GPCD: 33.33%; GPYA: 20.00%; GPQIN: 10.29%; GPQIO: 16.46%; and RP: 44.07%).

Notably, in the bacterial–phage interdomain networks (Figure 3 and Figure 4f), the keystone species of captive giant panda populations were primarily phages (GPCD: 100% and GPYA: 91.67%), with a predominant representation of Myoviridae (GPCD: 49.28% and GPYA: 66.67%) and Siphoviridae (GPCD: 33.33% and GPYA: 25.00%). Especially in the bacterial–phage network of GPCD, the co-occurrence network, and modular analysis, they revealed a key module, Module II (Figure 3 and Appendix A), which clustered all the keystone species in GPCD and was primarily composed of Siphoviridae, Myoviridae, and Pseudomonadaceae. Moreover, Pseudomonadaceae exhibited apparent positive interactions with these phages in Module II. In contrast, bacterial species constituted all keystone species within the other groups’ bacterial–phage networks (Figure 4f). Specifically, Firmicutes accounted for the major component of keystone species in CA (70.00%), OC (60.00%), GPQIN (100%), and RP (88.46%), Proteobacteria served as the main component of keystone species in GPQIO (100%) and GPXXL (100%), and Bacteroidetes represented the primary component of keystone species in HE (70.00%).

### 3.4. Topological Features of Bacterial–Phage, Bacterial, and Phage Networks

To deepen our understanding of gut microbial community interactions and structure–function assembly, we conducted a detailed analysis of the topological features and modularity of bacterial–phage, bacterial, and phage networks within each group (Table 2 and Appendix A). Cohesion serves as an indicator to quantify the strength of interactions among microorganisms. In the bacterial–phage interdomain networks (Figure 5a), we observed higher levels of cohesion in GPQIO, GPCD, and RP, while CA, HE, and GPXXL exhibited lower levels of cohesion. Specifically, GPQIO and GPCD demonstrated the highest positive and negative cohesion levels, respectively, with CA showing the lowest values for both positive and negative cohesion. Additionally, the average degree and modularity could further reflect the connectivity and complexity of the network. In the bacterial–phage interdomain networks, GPQIO displayed the highest average degree value and negative correlation rate (74.95%) (Table 2), suggesting that negative interactions may be predominant in its bacterial–phage interdomain network. In the bacterial networks (Figure 5b), GPQIO and CA exhibited the highest and lowest levels of cohesion (positive and negative), respectively. In the phage networks (Figure 5c), the populations of giant pandas and red pandas cohabiting in the same ecological environment of the Xiaoxiangling mountains exhibited large differences in cohesion levels. RP demonstrated relatively higher levels of cohesion (positive and negative), while GPXXL showed the lowest cohesion values (positive and negative). Furthermore, the modular analysis of the phage network revealed that GPXXL had the lowest degree of modularity (modularity value: 0.248). In addition, we conducted further investigation into the response of the cohesion level in the microbial network to different dietary preferences (Appendix A). The findings demonstrate an increasing trend in the total cohesion level of the gut bacterial, phage, and bacterial–phage interdomain networks, progressing from carnivorous (CA) to herbivorous (HE), then to omnivorous (OC) and finally to bamboo-eating species.

## 4. Discussion

### 4.1. Alpha Diversity and Composition Correlation of Gut Bacterial and Phage Communities in Mammals

As a type of virus that infects bacteria, phages are the most abundant members of all microorganisms, and they have been widely postulated to significantly outnumber their bacterial hosts, with an estimated ratio of approximately 10:1 within the gut [21,34,91,92]. Although indications support the potential of specific phages to infect multiple bacterial species simultaneously [46,93,94], the preponderance of evidence suggests that the majority of described phages exhibit specificity in invading only a few strains within the same bacterial species [95]. Despite the substantial numerical dominance of phages in the gut, the findings of this study reveal that the alpha diversity of bacterial communities in the mammalian gut surpasses that of phage communities in all statistically significant groups. In addition, a positive correlation in composition between gut bacterial and phage communities has been identified in non-human primates [29]. Similarly, our study demonstrated a positive correlation in composition between gut bacterial and phage communities across all studied bamboo-eating species (GPCD, GPYA, GPQIN, GPQIO, GPXXL, and RP), carnivores (CA), herbivores (HE), and omnivores (OC). A higher heterogeneity in the gut bacterial community composition was consistently accompanied by an elevated level of heterogeneity in the phage community composition [29]. These findings may further support the potential generalization of this relationship from primates to a wider range of other mammalian species.

Furthermore, previous studies in human twins have provided evidence suggesting a positive correlation between the alpha diversity of bacterial and phage communities in the gut, with a richer microbiome being associated with a richer phage community [62]. Genomic studies have revealed that the co-evolution of bacteria and phages plays a crucial driving role in molecular evolution and promotes the formation and maintenance of microbial diversity within the gut [47]. Bacterial–phage co-evolution is generally present in a wide range of ecosystems, including the gut, oceans, and soils, where highly diverse bacterial communities contribute to increased phage diversity [46,47]. The diversity of gut bacteria can expand the host range of phages by providing a variety of intermediate hosts. The P10 phage was able to select other intermediate bacterial hosts, inducing homologous intragenomic recombination and evolutionary events, thereby promoting the adaptability and long-term persistence in the mouse gut [46]. In turn, phages could drive bacterial diversity by maintaining the interactions and co-evolution with their bacterial hosts [96]. Diverse phages can attack various host bacteria, enhancing the potential for gene mutation and horizontal gene transfer, thereby expanding bacterial heterogeneity and evolution [36,97]. Horizontal gene transfer mediated by temperate bacteriophages can increase bacterial pathogenicity and adaptability by conferring or enhancing virulence to bacterial hosts, such as *Vibrio cholerae*, *Corynebacterium diphtheriae*, *Clostridium botulinum*, *Staphylococcus aureus*, *Streptococcus pyogenes*, and *Salmonella enterica* [41]. Our work revealed a higher diversity of mammalian gut bacterial communities and a positive bacterial–phage correlation in community composition, which may reflect a co-evolutionary potential of gut bacteria and phages generally present in a broader range of mammalian species in the natural world. This finding holds significant implications for a thorough understanding of the dynamic processes within the gut microbial ecosystem and provides insights for future research endeavors [98]. An in-depth exploration of the co-evolution between bacteria and phages is crucial for comprehending the mechanisms underlying microbial diversity maintenance and gut health [13,29] in mammals, particularly in endangered species. An improved comprehension of this co-evolutionary potential in mammals can guide the development of innovative intervention measures and conservation strategies aiming to safeguard the health of a broader range of mammalian species.

### 4.2. Diet and Phylogeny Influence the Structure of Mammalian Gut Bacterial and Phage Communities

Our work further revealed the significant effects of diet and phylogeny on both the gut bacterial and phage communities in mammals. We discovered that the impact of dietary factors on the gut bacterial and phage communities is more significant than that of phylogenetic factors at the order level but less significant than that of phylogenetic factors at the family level. However, merely considering the importance of a single influencing factor in isolation may limit a comprehensive understanding of the true dynamics, as the gut microbiome is actually regulated by multiple complex factors in a holistic manner [99,100]. It has been indicated that both the diet and host phylogeny exert an influence on the mammalian gut bacterial community, with gut bacterial diversity tending to increase from carnivores to omnivores and then to herbivores [28]. Similarly, our study further revealed the differential impact of dietary factors on the gut phage community across different taxonomic species. Specifically, the relative abundance of Podoviridae decreased from carnivores (CA: 89.53%) to herbivores (HE: 39.37%) to omnivores (OC: 1.11%), whereas Siphoviridae increased from carnivores (CA: 2.07%) to herbivores (HE: 15.70%) and then to omnivores (OC: 66.26%).

In addition, Beta diversity analyses have shown relatively small variation in both gut bacterial communities and phage communities between captive giant pandas (GPCD and GPYA) and OC (including Sun bear, brown bear, and Asian black bear). It has been reported that the same bamboo-eating behavior drove the similarity in gut microbiota composition between giant pandas (*Ailuropoda melanoleuca*) and red pandas (*Ailurus styani*) [101]. However, despite the specialized bamboo diet of giant pandas [102], their gut structure and microbial composition remain analogous to those of other bear species belonging to the family Ursidae [28,59,103]. Moreover, the diet of captive giant pandas is not solely composed of bamboo but also includes other food sources containing starch components, fruits, and other prepared foods [104], which may result in a dietary pattern that leans towards being weak-omnivorous rather than strictly bamboo-eating. Previous studies have reported that the significant bacterial composition in giant pandas, brown bears (*Ursus arctos*), and Asian black bears (*Ursus thibetanus*) consists of Proteobacteria and Firmicutes [59,105,106]. Therefore, we hypothesize that the closer phylogenetic relationship between giant pandas and bears, as well as the omnivorous dietary habits of captive giant pandas, may contribute to the higher similarity of gut bacterial and phage communities between captive giant panda populations (GPCD and GPYA) and omnivorous animals (OC) in this study.

### 4.3. Interactions between Gut Bacterial and Phage Communities

In contrast to the preliminary insights and contributions offered by diversity and composition analyses in characterizing microbial communities [52,83,107], network analyses present an opportunity to delve deeper into the intricate dynamics of microbial ecosystems [108], which provide a distinct perspective to facilitating comprehensive investigations into microbial communities by unveiling the interactions, resource utilization patterns, ecological assembly rules, and other essential aspects among microbial communities [52,109,110]. Complex interactions among microorganisms profoundly impact the functioning and stability of entire microbial ecosystems [83,108,111]. The cohesion metrics in co-occurrence network analysis can be used to reflect the strength of interactions of microbial communities [88,89]. The positive and negative cohesion value increase indicates enhanced positive (synergistic and symbiotic) and negative microbial interactions (competitive, antagonistic, and parasitic), respectively. Meanwhile, total cohesion, which is the sum of the absolute values of positive and negative cohesion values, can offer a comprehensive overview of the degree of interplay among microorganisms [88]. This study employed network analysis to reveal the interactions between gut bacterial and phage communities in different mammals. An increase in total cohesion levels was observed from carnivores (CA) to herbivores (HE), then to omnivores (OC), and finally to bamboo-eating species (GPCD, GPYA, GPQIN, GPQIO, GPXXL, and RP). This suggests a stronger interaction between the bacterial and bacteriophage communities in the gut of bamboo-eating species compared to other mammals. Phages typically interact specifically with individual bacterial strains, coexisting in a complex and modular interaction network [112]. As bacterial predators, phages can regulate bacterial abundance and stability, and their interactions may directly impact host health and disease [112]. The influence of intricate external environments on phage–bacterial interactions remains unpredictable. It has been shown that certain factors such as diet, lifestyle, and health status may not significantly affect marker phages (CrAss-phages) in the human gut [113,114]. Moreover, additional experimental evidence suggests that compounds such as bile salts and sodium dodecyl sulfate can diminish the inhibitory effects of phages on bacterial growth [115], while antibiotics may enhance interactions between host gut bacteria and phages, resulting in a more closely interconnected gene exchange network [112]. We observed that different dietary preferences affect the interactions between bacteria and phages in the gut of mammalian hosts, which further strengthens the study of the roles of bacteria and phages in different mammalian ecosystems, and contributes to the development of conservation strategies tailored to different endangered animal populations. In addition, the cohesion level manifested in microbial communities could indirectly reflect the resistance to external environmental disturbances and the stability of microbial communities [109,110]. Microbial communities with greater absolute values of negative cohesion were likely to be more stable [90,116], as they exhibit strong competitive interactions, resulting in relatively smaller variations in microbial composition caused by external factors. Consequently, we hypothesize that the bacterial–phage network in bamboo-eating species with the highest level of negative cohesion exhibits greater stability and resistance to external environmental disturbances (Appendix A).

Different microbial nodes perform distinct topological functions within the network [54,117], enabling the improved identification of functional assembly patterns within microbial communities through the partitioning of closely associated nodes into modules [52,80,83,109,118]. Furthermore, microbial network analyses in various ecological environments, including the gut, water, plants, and soil, have consistently demonstrated that keystone species could play a critical role in maintaining the stability of microbial communities and facilitating efficient material and energy cycling by serving as the central hub for the structure and functionality of the entire network [52,80,119,120,121]. Notably, we observed that the keystone species in the gut bacterial–phage network of captive giant pandas mainly consisted of phages (GPCD: 100% and GPYA: 91.67%), whereas the keystone species in other groups were exclusively bacteria. Moreover, all keystone species in a bacterial–phage network of GPCD were found to be clustered in Module II, which exhibited obvious positive interactions between the predominant bacteriophages (Siphoviridae and Myoviridae) and bacteria (Pseudomonadaceae). These findings may indicate that phages play a more vital functional role in maintaining the coexistence patterns and interactions between gut bacteria and phages in captive giant pandas. In addition, the significant impact of captivity on the gut bacterial community structure and functionality of wild endangered animals has been widely reported. Furthermore, the living environment is also a crucial factor influencing the host gut phage community [59,122]. Limited previous research evidence in species such as vulture (*Gyps hinalayensis*) [123], Tasmanian devil (*Sarcophilus harrisii*) [124], and giant panda (*Ailuropoda melanoleuca*) [59] have indicated that controlled captive environments (e.g., dietary changes, microbial transmission, and human interference) can alter the composition and diversity of the host gut phage community. We speculate that the unique captive environment of giant pandas [4,16,52,61,104,125,126], characterized by factors such as a high-starch and -milkfat diet, increased antibiotic usage, and frequent human contact, may contribute in part to the pivotal role of phages in shaping the coexistence pattern of the gut bacterial and phage communities in captive giant pandas.

### 4.4. Potential Correlations of Gut Phages with Corresponding Bacterial Hosts

The complex interactions between gut bacteria and phages constitute a fundamental inquiry in the study of the gut microbiome [95,114,127,128]. As top predators of bacteria, phages play a role in regulating the gut bacterial populations; some clues may indicate a possible correlation between heightened levels of specific phages and the decline of particular bacterial taxa in the gut [95,129,130]. In addition, the “Piggyback-the-Winner” theory proposes that when host bacteria exhibit high abundance and growth rates, phages may choose a symbiotic strategy of integrating their genomes into the host rather than killing the host [39]. It has been reported that the dynamics of gut phage in giant pandas follow this theory [61]. Specifically, when Enterobacteria were abundant, a higher abundance of corresponding phages, such as Escherichia and Enterobacteria phages, was observed. This is also consistent with some of the results observed in this study. In GPXXL, *Pseudomonas* (46.16%) and its corresponding phage, 201phi2-1 (59.35%) (Appendix A) were identified as the predominant constituents within the gut bacterial and phage communities, respectively. Several studies have reported the propagation and isolation [131], nucleus-like assembly [132], and protein maturation processes [133] of *Pseudomonas* phage 201phi2-1. *Pseudomonas* phage 201phi2-1 is a jumbo phage that encodes the protein gp105, which plays a significant role in the formation of the nucleus-like compartment during the infection of *Pseudomonas chlororaphis* by phage 201phi2-1 [132]. Similarly, in CA, Bacteroidaceae (32.48%) were identified as the most abundant constituent in the gut bacterial community, while crAssphage (crAssphage cr110_1: 43.58% and crAssphage cr11_1: 36.66%) exhibited an overwhelming dominance in the gut phage community (Appendix A). CrAssphage, primarily parasitizing diverse bacterial species within the Bacteroidetes in the intestinal tract [134,135,136], represents the predominant phage type within the human gut [137,138,139]. It has been extensively used as a robust indicator for characterizing the human gut virome and assessing fecal water contamination [140,141,142]. In sharp contrast, crAssphage is seldomly detected in animal feces [140], with reports currently confined to non-human primates [113], pigs [143], and cats [134] exclusively. It is noteworthy that our findings highlight an exceptionally abundant crAssphage composition in the gut of carnivorous mammals (CA), thereby significantly expanding the knowledge base surrounding crAssphage within a broader spectrum of mammalian species.

### 4.5. The Uniqueness of the Gut Bacterial and Phage Community in Qinling Giant Pandas

Previous studies have shown variations in gut microbiome’s structural composition among giant pandas’ different geographic populations [52,144,145]. Compared to non-Qinling populations of giant pandas, Qinling giant pandas typically exhibit more pronounced differences in their gut microbial structure. A recent study has revealed that three enterotypes, *Escherichia* (captive period), *Clostridium* (reintroduction training period), and *Pseudomonas* (wild period), could characterize the adaptive evolution of gut microbes in captive giant pandas that have been reintroduced to the wild [146]. We found that the gut microbiota of the Qinling giant panda population preferred the gut enterotypes of the reintroduction training period, as *Clostridium* (38.99%) was the most dominant bacterium in GPQIN, while *Escherichia* and *Pseudomonas* were the most abundant constituents in captive (GPCD: 29.09 and GPYA: 37.00%) and wild (GPQIO: 18.56% and GPXXL: 44.16%) giant pandas, respectively. In addition, previous studies have reported that Qinling populations exhibit a higher abundance of *Clostridium* and vancomycin resistance genes [144] and a lower virulence level of *P. aeruginosa* [145]. Here, our work further revealed the unique composition pattern of gut phages in Qinling giant panda population, where GPQIN exhibited the highest relative abundance of Drexlerviridae (74.83%) phages compared to other giant panda populations. These findings further expand our understanding of gut phage variations across geographically distinct giant panda populations, providing valuable insights for the ecological health and conservation strategies tailored for these iconic species.

## 5. Conclusions

In summary, this study investigated the relationships (diversity and composition) and potential interactions between gut bacterial and phage communities in mammals with different dietary preferences (bamboo-eating, carnivorous, omnivorous, and herbivorous), along with examining the impacts of diet and phylogeny on these communities. We found a higher alpha diversity in gut bacterial than in phage communities and identified positive correlations between bacterial and phage community compositions, which may indicate that the higher co-evolutionary potential of bacteria and phages is generally present in a broad range of mammalian species. Moreover, we uncovered significant effects of both dietary and phylogenetic factors on the gut bacterial and phage communities. Specifically, the impact of phylogenetic factors at the family level were more pronounced on these communities than dietary factors and phylogenetic factors at the order level. In this work, the higher similarity observed in gut bacterial communities and phage communities between captive giant panda populations (GPCD and GPYA) and omnivorous animals (OC; including Sun bear, brown bear, and Asian black bear), compared to other groups, may be attributed to similar omnivorous dietary preferences and closer phylogenetic relationships (family Ursidae). In addition, we identified specific taxonomic phages that demonstrate noteworthy distribution patterns. Specifically, crAssphage bacteriophages, serving as indicators of human gut viral communities and fecal water pollution, were prevalent in carnivorous animal populations (CA). Moreover, when compared to other giant panda populations, Drexlerviridae phages exhibited a significant predominance in the Qinling giant panda population.

Our analysis of the bacterial–phage interdomain network revealed potential connections and interactions between bacterial and phage communities in the mammalian gut. Notably, bamboo-eating species exhibited more substantial interactions between gut bacteria and phages than other mammalian taxa, including carnivores, herbivores, and omnivores. Moreover, keystone species analyses suggest that phage communities may be crucial in the steady state and interactions between gut bacteria and phages in captive giant pandas (GPCA and GPYA) compared to gut bacterial communities. Notably, despite the ability of network analysis to provide clues about the overall assembly rules and potential interactions within microbial communities based on data correlations, it cannot directly prove the authenticity of these interactions in actual microbial ecosystems [147,148,149,150,151]. Nonetheless, it indicates a reasonable direction for future research by filtering out low-probability scientific hypotheses [147]. In the future, experimental validation can be conducted by incorporating more samples and more efficient experimental designs, thereby transforming the possibilities offered by network analysis into actual evidence.

## Figures and Tables

**Figure 1 microorganisms-12-00713-f001:**
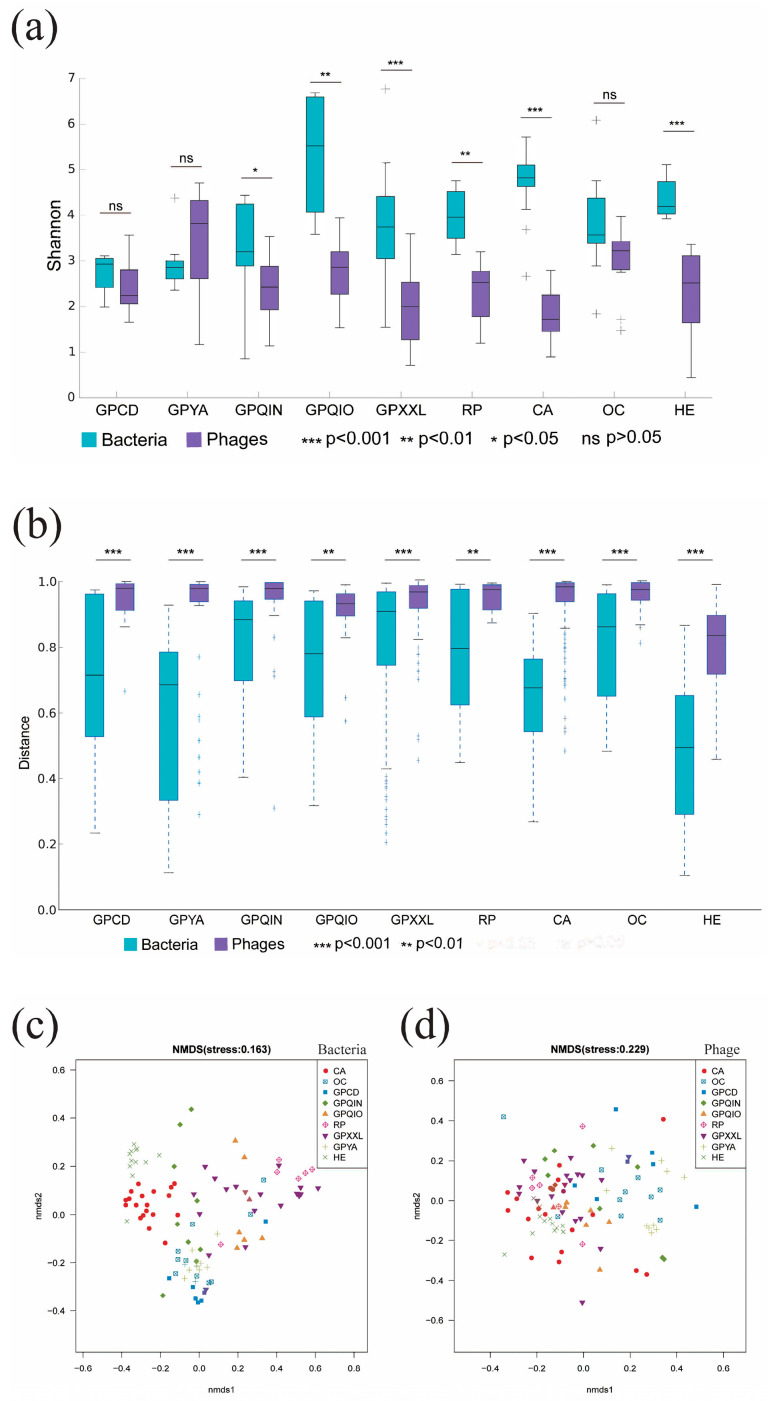
Diversity analysis of gut bacterial and phage communities. (**a**) Alpha diversity analysis of gut bacterial and phage composition within each group. (**b**) Beta diversity analysis of gut bacterial and phage composition within each group. (**c**) The non-metric multidimensional scaling (NMDS) analysis of the compositions of gut bacteria among all groups (Adonis, *R*^2^ = 0.39781, *p* = 0.001). (**d**) The non-metric multidimensional scaling (NMDS) analysis of the compositions of gut phage among all groups (Adonis, *R*^2^ = 0.16089, *p* = 0.001). GPCD, captivity giant panda in Chengdu. GPYA, captivity giant panda in Ya’an. GPQIN, wild giant panda in Qinling. GPQIO, wild giant panda in Qionglai. GPXXL, wild giant panda in Xiaoxiangling. RP, wild red panda in Xiaoxiangling. CA, carnivorous mammal. OC, omnivorous mammal. HE, herbivorous mammal.

**Figure 2 microorganisms-12-00713-f002:**
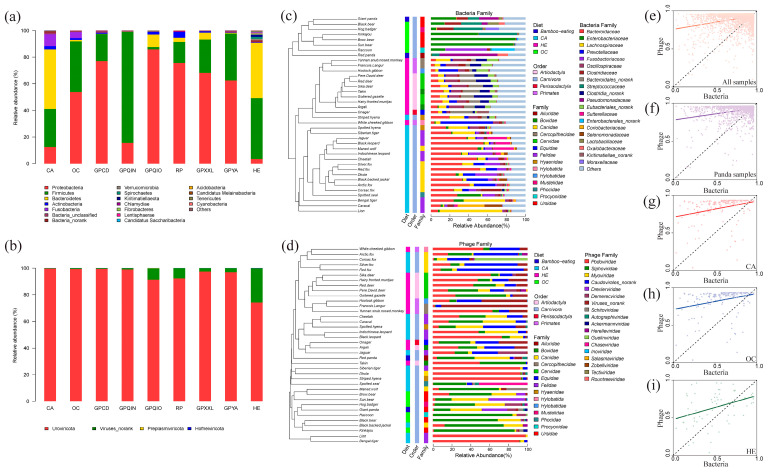
Compositional analysis of gut bacterial and phage communities. (**a**) Composition of the gut bacterial community at the gate level. (**b**) Composition of the gut phage community at the phylum level. (**c**) Composition of the gut bacterial community at the family level. (**d**) Composition of the gut phage community at the family level. The relationship of compositions between the gut bacterial community and phage community based on Sørensen’s dissimilarity in (**e**) all samples (Slope: 0.1268, *p*-value: 2.2 × 10^−16^), (**f**) bamboo-eating panda samples (GPCD, GPYA, GPQIN, GPQIO, GPXXL, and RP; Slope: 0.1268, *p*-value: 2.2 × 10^−16^), (**g**) carnivore samples (CA; Slope: 0.1780, *p*-value: 0.0002), (**h**) omnivore samples (OC; Slope: 0.1779, *p*-value: 0.0002), and (**i**) herbivore samples (HE; Slope: 0.2225, *p*-value: 0.0022). We are applying the linear model fit (solid line) to explain the relationship between the two microbial communities.

**Figure 3 microorganisms-12-00713-f003:**
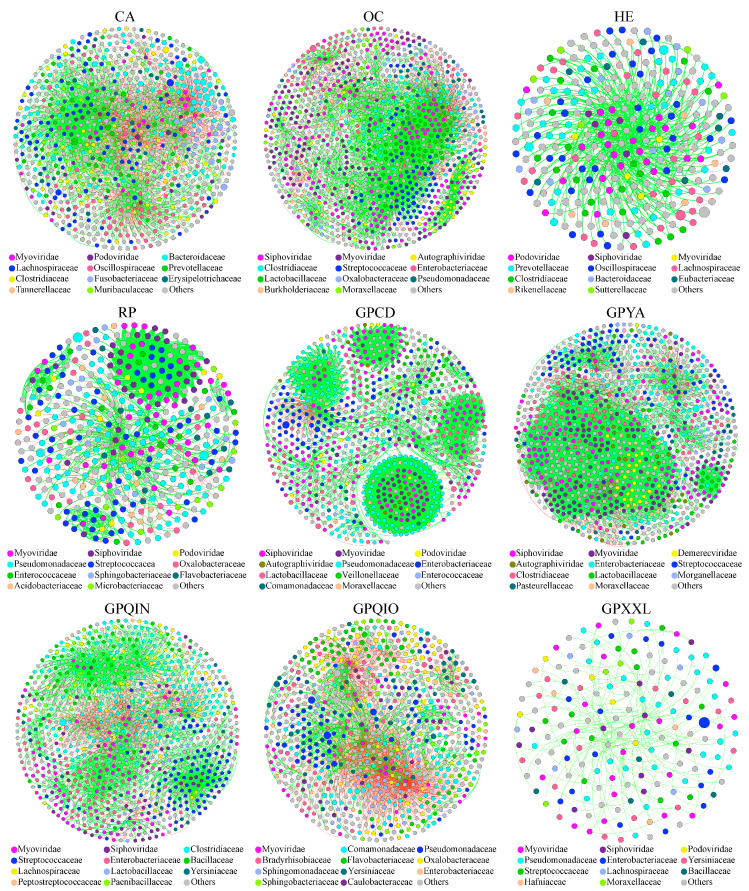
Co-occurrence network analysis of bacterial–phage interdomain communities in all groups. The nodes representing taxonomic groups of bacteria and phages are color-coded based on their respective family in bacterial–phage networks. The size of the node represents the number of connections (degree value). Different colored edges represent positive (green) and negative (red) correlations between nodes. GPCD, captivity giant panda in Chengdu. GPYA, captivity giant panda in Ya’an. GPQIN, wild giant panda in Qinling. GPQIO, wild giant panda in Qionglai. GPXXL, wild giant panda in Xiaoxiangling. RP, wild red panda in Xiaoxiangling. CA, carnivorous mammal. OC, omnivorous mammal. HE, herbivorous mammal.

**Figure 4 microorganisms-12-00713-f004:**
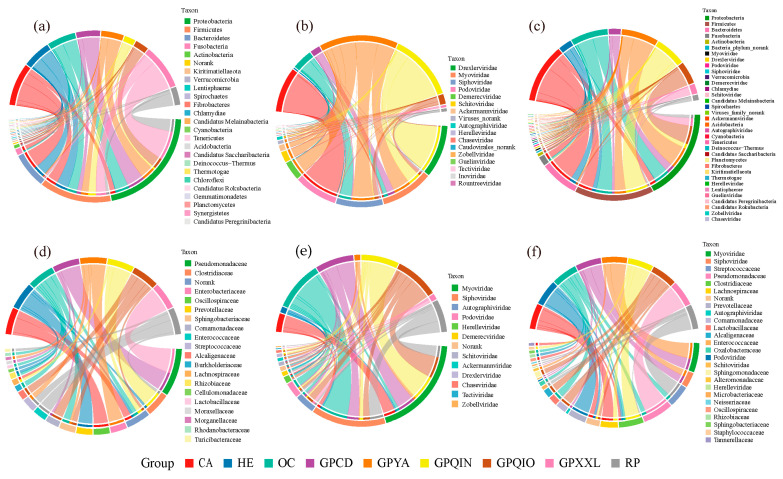
Information on the node compositions at the family level within the networks of bacterial, phage, and bacterial–phage communities in all groups. The compositions of all nodes within the networks of bacterial (**a**), phage (**b**), and bacterial–phage (**c**) communities. The compositions of keystone species (nodes) within the networks of bacterial (**d**), phage (**e**), and bacterial–phage (**f**) communities. On the outer ring, different colors represent distinct groups (upper half of the chord diagram) and the composition of categories at the family level (lower half of the chord diagram). A specific ribbon color represents each group, and the width of each ribbon indicates the abundance of each family within each group. The labels explaining the groups are positioned at the bottom of the figure, while the labels describing the composition of categories are placed on the right side of each chord diagram. CA, carnivorous mammal. HE, herbivorous mammal. OC, omnivorous mammal. GPCD, captivity giant panda in Chengdu. GPYA, captivity giant panda in Ya’an. GPQIN, wild giant panda in Qinling. GPQIO, wild giant panda in Qionglai. GPXXL, wild giant panda in Xiaoxiangling. RP, wild red panda in Xiaoxiangling.

**Figure 5 microorganisms-12-00713-f005:**
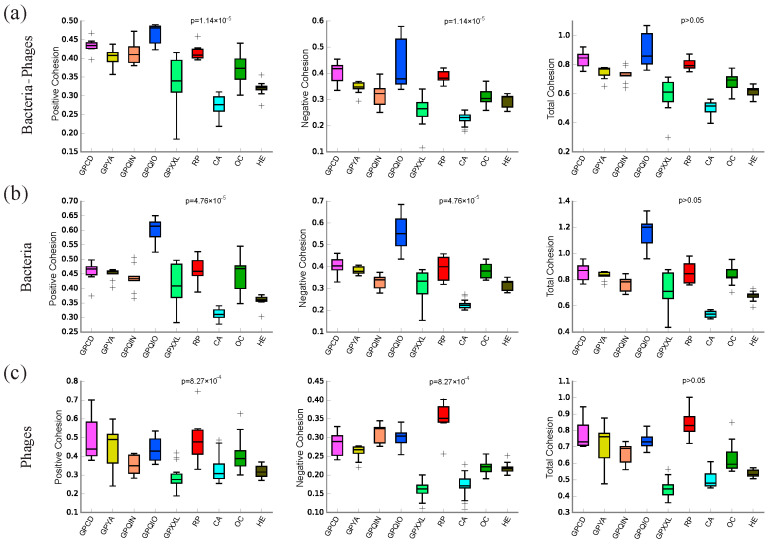
The cohesion levels of gut bacterial–phage, bacterial, and phage communities in all groups. The positive cohesion, negative cohesion, and total cohesion within (**a**) bacterial–phage communities, (**b**) bacterial communities, and (**c**) phage communities in the gut of each group. Outliers are indicated by the symbol “+”. GPCD, captivity giant panda in Chengdu. GPYA, captivity giant panda in Ya’an. GPQIN, wild giant panda in Qinling. GPQIO, wild giant panda in Qionglai. GPXXL, wild giant panda in Xiaoxiangling. RP, wild red panda in Xiaoxiangling. CA, carnivorous mammal. OC, omnivorous mammal. HE, herbivorous mammal.

**Table 1 microorganisms-12-00713-t001:** Testing the statistical significance of diet and phylogeny (at the levels of order and family) on host gut bacterial and phage communities through the analysis of similarity (ADONIS) statistics.

Group	Type	Df	F	*R* ^2^	*p*
Bacteria	Diet	3	9.4758	0.45537	0.001
	Family	3	4.6384	0.7153	0.001
	Order	3	4.7941	0.29726	0.001
Phages	Diet	3	2.9126	0.16211	0.001
	Family	3	1.4877	0.44625	0.001
	Order	3	1.9018	0.14369	0.001

**Table 2 microorganisms-12-00713-t002:** Topological properties of bacterial, phage, and bacterial–phage interdomain networks in all groups. CA, carnivorous mammal. HE, herbivorous mammal. OC, omnivorous mammal. GPCD, captivity giant panda in Chengdu. GPYA, captivity giant panda in Ya’an. GPQIN, wild giant panda in Qinling. GPQIO, wild giant panda in Qionglai. GPXXL, wild giant panda in Xiaoxiangling. RP, wild red panda in Xiaoxiangling.

Network	Group	Node	Edge	Positive Correlation	Negative Correlation	Positive Correlation Rate	Negative Correlation Rate	Average Degree	Modularity Value
Bacteria	CA	859	8135	7880	255	96.87%	3.13%	18.94	0.627
	HE	717	1846	1634	212	88.52%	11.48%	27.8	0.766
	OC	861	7850	7459	391	95.02%	4.98%	103.92	0.71
	GPCD	983	16,533	13,698	2835	82.85%	17.15%	83.25	0.708
	GPYA	656	2402	2292	110	95.42%	4.58%	47.02	0.843
	GPQIN	373	522	518	4	99.23%	0.77%	111.91	0.925
	GPQIO	646	6135	5461	674	89.01%	10.99%	471.74	0.733
	GPXXL	797	9882	9882	0	100.00%	0.00%	73.41	0.676
	RP	963	7041	5394	1647	76.61%	23.39%	83.39	0.857
Phages	CA	273	1699	1699	0	100.00%	0.00%	12.45	0.845
	HE	98	200	200	0	100.00%	0.00%	4.74	0.88
	OC	615	10,966	10,966	0	100.00%	0.00%	43.31	0.76
	GPCD	291	5534	5532	2	99.96%	0.04%	38.91	0.715
	GPYA	435	6167	6167	0	100.00%	0.00%	60.09	0.695
	GPQIN	282	3487	3487	0	100.00%	0.00%	28.12	0.524
	GPQIO	373	6550	6550	0	100.00%	0.00%	40.23	0.653
	GPXXL	386	9544	9544	0	100.00%	0.00%	50.46	0.248
	RP	245	4141	4138	3	99.93%	0.07%	35.16	0.667
Bacteria–Phages	CA	846	2598	1210	1388	46.57%	53.43%	134.61	0.52
	HE	284	353	316	37	89.52%	10.48%	28.4	0.772
	OC	1146	4181	3434	747	82.13%	17.87%	83.8	0.606
	GPCD	864	8406	8161	245	97.09%	2.91%	59.63	0.516
	GPYA	1008	9423	8331	1093	88.40%	11.60%	70.33	0.463
	GPQIN	967	3054	2085	969	68.27%	31.73%	67.41	0.667
	GPQIO	840	2104	527	1577	25.05%	74.95%	204.07	0.553
	GPXXL	177	173	173	0	100.00%	0.00%	40.19	0.867
	RP	418	1515	1335	180	88.12%	11.88%	23.78	0.585

## Data Availability

Data are contained within the article (and Appendix A).

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
