# Peer review of "The Potential of Co-Evolution and Interactions of Gut Bacteria–Phages in Bamboo-Eating Pandas: Insights from Dietary Preference-Based Metagenomic Analysis"

_microorganisms, 2024, doi:10.3390/microorganisms12040713_

Round 1
Reviewer 1 Report
Comments and Suggestions for Authors
In the article, authors systematically reviewed and used published data to find gut communities members interactions between different mammal species, concentrated on two panda species (~50% of all species). The heterogeneity of the data used is, on the one hand, very interesting from the point of view of data generalization, on the other hand it creates some problems with their interpretation, as different host species of metagenomes start to have different weights. Perhaps, this methodological problem does not have unambiguous resolutions due to the insufficiency of metagenomic data, but it is worth considering when interpreting them.
Considering the fact that there are significant differences in alpha-diversity calculating by Shannon index, it would be also interesting to measure dominance and evenness metrics and estimate differences not only between groups but also between samples.
Major article part is dedicated to correlations determination. However, the number of samples in each group is really small, so some correlations should be false-significant, and sample size considerably influences results. Authors applied threshold to the data before visualizing networks, but maybe correction to p-value or sample size itself also will be worth it. For example, not many significant correlations in the GPXXL group might relate to the amount of metagenomes. Maybe this remark is irrelevant because of the provided Table S3, mentioned in the methods section.
Unfortunately, I believe that due to technical problems I did not have access to supplementary materials, so I do not have relevant information about some study results and data collection information. Also, MDPI slightly compresses images, so I had problems with some of them (especially, figure 2 c and d). So, please update them in higher quality in the current version of the article
There are some minor observations:
-
Title contains a typo “-”, and in my opinion is slightly irrelevant. Authors have performed really big work not only on pandas, but also on other mammalian species, and have identified not only Bacteria-Phages interactions, but systematized composition and connection information in groups too. Therefore, phylogenetic analysis of host-microbiota interactions were not performed. Perhaps the authors should edit the title to better reflect the content of the article.
-
The introduction focuses on some general considerations about the gut microbiome. Much of the article focuses on measurements of interactions within them. It would have been useful to write more about previous network studies.
-
Most often figures and tables are placed after they are mentioned. So it might be worth relocating them.
-
Line 62, 91 and other: space missed between sentence and citation, e.g. in “phages[29]”
-
Line 274-277: this can be proved using PCoA, but it is not very necessary.
-
Line 371-374: I am not sure how to interpret “positive correlations in compositions”. Does this mean that if there are more bacteria getting bigger, all phages are getting bigger in about the same ratios in all groups? It might be worth clarifying this sentence, right now it is too general.
The article submitted by the authors is of high quality and I found it very interesting and enjoyable to review.
Author Response
Comment:
Title contains a typo “-”, and in my opinion is slightly irrelevant. Authors have performed really big work not only on pandas, but also on other mammalian species, and have identified not only Bacteria-Phages interactions, but systematized composition and connection information in groups too. Therefore, phylogenetic analysis of host-microbiota interactions were not performed. Perhaps the authors should edit the title to better reflect the content of the article.
Replay: Thank you for your valuable suggestion. We have diligently rectified the notation errors and also reevaluated and revised the title (Line 2-4) to more accurately represent the content of the article.
The introduction focuses on some general considerations about the gut microbiome. Much of the article focuses on measurements of interactions within them. It would have been useful to write more about previous network studies.
Replay: As recommended by the reviewer, we have included additional content on previous network studies to enhance the introduction section (Lines 80-103).
MDPI slightly compresses images, so I had problems with some of them (especially, figure 2 c and d). So, please update them in higher quality in the current version of the article
Most often figures and tables are placed after they are mentioned. So it might be worth relocating them.
Replay: We believe this is an excellent suggestion for enhancing our manuscript. We have enhanced the quality (300 dpi) of all the images in the manuscript and placed the figures and tables after they are referenced (Lines 231-240, 249-251, 275-285, 304-311, 312-325, 380-393).
Line 62, 91 and other: space missed between sentence and citation, e.g. in “phages [29]”
Replay: We were really sorry for our careless mistakes. Thank you for pointing out the error. We have rectified the issue (missing space between sentence and citation) in the manuscript, and all modifications have been highlighted in red within the document, such as "Line 69, phages [41]", "Line 417, foods [106]", and "Line 546, theory [127]".
Line 274-277: this can be proved using PCoA, but it is not very necessary.
Replay: Thank you for your suggestion. We have reviewed the point you raised about not conducting PCoA analysis. In light of this, we believe that PCoA analysis is indeed not necessary for this study, and as such, we have chosen to maintain our original analytical approach. We hope this explanation meets your expectations.
Line 371-374: I am not sure how to interpret “positive correlations in compositions”. Does this mean that if there are more bacteria getting bigger, all phages are getting bigger in about the same ratios in all groups? It might be worth clarifying this sentence, right now it is too general.
Replay: Thank you very much for the reviewer's suggestion, and considering that the current statement on "positive correlations in compositions" is indeed too general, we have further clarified and elucidated it (Line 407-412, A higher heterogeneity in the gut bacterial community composition was consistently ac-companied by an elevated level of heterogeneity in the phage community composition). Thank you again for your positive comments and valuable suggestions to improve the quality of our manuscript.
Reviewer 2 Report
Comments and Suggestions for Authors
The authors in their study "The Potential of Co-evolution and Interactions of Gut Bacteria-Phages in bamboo-eating pandas: Insights from Diet and Phy-logeny-Based Metagenomic Analysis" investigate the interaction between the gut mi-crobiota and phages in bamboo-eating pandas
1- I think the study is well designed and most of the sections of the manuscript are adequately described or presented except the discussion which requires further improvement to explain the findings (interactions between phages and microbiota) and link to healthy lifestyle or diet style of the animal
2- The language requires further improvement
Comments on the Quality of English LanguageModerate editing of English language required
Author Response
Comment:
I think the study is well designed and most of the sections of the manuscript are adequately described or presented except the discussion which requires further improvement to explain the findings (interactions between phages and microbiota) and link to healthy lifestyle or diet style of the animal
Replay: Thank you for your positive feedback on our study and valuable suggestions. We have revised the discussion section (Line 508-522) based on your suggestions to further expand the discussion of gut phage-bacteria interactions in this study. We believe that these additions will enhance the depth and relevance of the article, helping readers better understand the significance of the research and its potential applications.
The language requires further improvement.
Moderate editing of English language required
Replay: Thank you for your feedback. We acknowledge the need for further improvement in the language of the manuscript. We will work on moderate editing of the English language to enhance the clarity and overall quality of the text. We have incorporated this advice by revising the sentences, maintaining the original content and structure of the paper. Although we have not listed all the changes, they are highlighted in red in the revised manuscript. Your suggestions are valuable to us, and we are committed to enhancing the language to ensure better readability and understanding of the study.
Reviewer 3 Report
Comments and Suggestions for Authors
Overall, the abstract effectively communicates the study's objectives, methods, and key findings, providing a solid foundation for the importance of the research in the context of mammalian gut microbiota and phage communities.
Specify the time frame or publication years of the metagenomes to provide context for the data collection. Clarify how the metagenomes from different geographic populations were selected or what criteria were used for inclusion.
Consider breaking down the complex sentences into shorter ones to improve readability.
Use bullet points or subheadings to clearly delineate different steps in the metagenomic analysis process, making it easier for readers to follow.
Clarify the rationale behind integrating an additional set of 41 metagenomes, including the distribution across carnivores, herbivores, and omnivores.
In the statistical analysis section, briefly explain the rationale and relevance of the methods employed, such as the Adonis test and Sørensen's dissimilarity.
Clarify the significance of the positive correlation between gut bacterial and phage communities in the context of the study's objectives.
Emphasize any unexpected or novel findings that may contribute to the broader understanding of the gut microbiome.
Consider mentioning specific examples or applications of co-evolution in other ecosystems or species for a more comprehensive discussion.
Provide a brief summary or synthesis of the observed impacts of diet and phylogeny on gut bacterial and phage communities in mammals.
Highlight any unexpected or noteworthy patterns, such as the differential impact of dietary factors at various taxonomic levels.
The conclusion provides a comprehensive summary of the study's key findings.
Reiterate the main findings in a concise manner at the beginning of the conclusion for emphasis.
Author Response
Comment:
Specify the time frame or publication years of the metagenomes to provide context for the data collection. Clarify how the metagenomes from different geographic populations were selected or what criteria were used for inclusion.
Clarify the rationale behind integrating an additional set of 41 metagenomes, including the distribution across carnivores, herbivores, and omnivores.
Replay: Thank you for your positive comments and valuable suggestions to improve the quality of our manuscript. Considering the Reviewer’s suggestion, we have made revision to strengthen the persuasiveness of our results. Specifically, we have included more information and improvements in the manuscript. For instance, details of the metagenomic data used have been updated in Supplementary information Table S1, including the publication years as recommended by the reviewer. Additionally, we have clarified the reasoning behind integrating the additional 41 metagenomes in the manuscript (Lines 104-130).
Consider breaking down the complex sentences into shorter ones to improve readability.
Replay: Thank you for your valuable feedback on the complexity of the sentences in the manuscript. I appreciate your suggestion to simplify the sentence structures by breaking down complex sentences into shorter ones to improve readability. We have implemented this advice by revising the sentences while ensuring that the content and structure of the paper remain unaffected. Although we have not listed all the changes, they are highlighted in red in the revised paper. We sincerely value the diligent work of Editors/Reviewers and hope that these corrections will be satisfactory. Through the use of shorter and more concise sentences, I believe the manuscript's readability and clarity have been
Use bullet points or subheadings to clearly delineate different steps in the metagenomic analysis process, making it easier for readers to follow.
In the statistical analysis section, briefly explain the rationale and relevance of the methods employed, such as the Adonis test and Sørensen's dissimilarity.
Replay: We consider this to be an excellent suggestion for enhancing the quality of the manuscript. In response to the reviewers' feedback on the "Methods and Materials" section, we have thoroughly reviewed and revised the text. We have incorporated subheadings to explicitly outline the various stages in the metagenomic analysis process (Lines 146-163), and have added the rationale and relevance of the statistical analysis methods used in this manuscript, including the Adonis test and the Sørensen's dissimilarity (Line165-182).
Clarify the significance of the positive correlation between gut bacterial and phage communities in the context of the study's objectives.
Consider mentioning specific examples or applications of co-evolution in other ecosystems or species for a more comprehensive discussion.
Replay: We would like to express our sincere appreciation for your professional review of our article. Thanks to your insightful suggestions, we have incorporated appropriate changes into the manuscript (Lines 397-444). The increased diversity of gut bacteria in mammalian taxa with varying dietary preferences, coupled with the observed positive correlation between gut bacteria and phages, further strengthens the argument for an enhanced coevolutionary potential between mammalian gut bacteria and phages. In accordance with your recommendations, we have extensively expanded the discussion on the coevolution of gut bacteria and phages by incorporating examples from previous studies on bacterial-phage coevolution across various ecosystems or species (Lines 420-422, 431-433).
Emphasize any unexpected or novel findings that may contribute to the broader understanding of the gut microbiome.
Provide a brief summary or synthesis of the observed impacts of diet and phylogeny on gut bacterial and phage communities in mammals.
Highlight any unexpected or noteworthy patterns, such as the differential impact of dietary factors at various taxonomic levels.
The conclusion provides a comprehensive summary of the study's key findings.
Reiterate the main findings in a concise manner at the beginning of the conclusion for emphasis.
Replay: We deeply value your constructive feedback and suggestions on our manuscript. We have meticulously reviewed your comments and implemented substantial revisions in the "Conclusions" section (Lines 590-624). This includes succinctly outlining the main findings at the outset of the conclusions (Lines 590-593), providing a comprehensive summary of the impacts of diet and phylogeny on the bacterial and phage communities in the mammalian gut (Lines 597-605), and emphasizing unexpected and innovative discoveries that contribute to a more profound understanding of the gut microbiome (Lines 605-610).
Your insightful guidance has been instrumental in refining the content, and I am confident that these adjustments have elevated the overall quality of the document. Thank you once again for your invaluable feedback.